# Total neoadjuvant therapy versus standard therapy in locally advanced rectal cancer: A systematic review and meta-analysis of 15 trials

Xiping Zhang[1], Shujie Ma[2], Yinyin Guo[3], Yang Luo[4], Laiyuan Li[5]*

**1** Department of General Surgery, Qinan Hospital, Tianshui, China, **2** Department of General Surgery, People's Hospital of Gannan, Hezuo, China, **3** Department of Pharmacy, Lanzhou University Second Hospital, Lanzhou, China, **4** Department of Neurology, The First Hospital of Lanzhou University, Lanzhou, China, **5** Department of Anorectal Surgery, Gansu Provincial Hospital, Lanzhou, China

* lilaiyuansysu@163.com

**Data Availability Statement:** All relevant data are within the paper.

**Funding:** the Natural Science Foundation of Gansu Province (No.20JR10RA371, L.LY), Fundamental

## Abstract

### Background

Neoadjuvant chemoradiotherapy (nCRT) before total mesorectal excision (TME) and followed systemic chemotherapy is widely accepted as the standard therapy for locally advanced rectal cancer (LARC). This meta-analysis was to evaluate the current evidence regarding nCRT in combination with induction or consolidation chemotherapy for rectal cancer in terms of oncological outcomes.

### Methods

A systematic search of medical databases (PubMed, EMBASE and Cochrane Library) was conducted up to the end of July 1, 2021. This meta-analysis was performed to evaluate the efficacy of TNT in terms of pathological complete remission (pCR), nCRT or surgical complications, R0 resection, local recurrence, distant metastasis, disease-free survival (DFS) and overall survival (OS) in LARC.

### Results

Eight nRCTs and 7 RCTs, including 3579 patients were included in the meta-analysis. The rate of pCR was significantly higher in the TNT group than in the nCRT group, (OR 1.85, 95% CI 1.39–2.46, p < 0.0001), DFS (HR 0.80, 95% CI 0.69–0.92, p = 0.001), OS (HR 0.75, 95% CI 0.62–0.89, p = 0.002), nCRT complications (OR 1.05, 95% CI 0.77–1.44, p = 0.75), surgical complications (OR 1.02, 95% CI 0.83–1.26, p = 0.83), local recurrence (OR 1.82, 95% CI 0.95–3.49, p = 0.07), distant metastasis (OR 0.77, 95% CI 0.58–1.03, p = 0.08) did not differ significantly between the TNT and nCRT groups.

### Conclusion

TNT appears to have advantages over standard therapy for LARC in terms of pCR, R0 resection, DFS, and OS, with comparable nCRT and postoperative complications, and no increase in local recurrence and distant metastasis.

Research Funds for the Central Universities (No. lzujbky-2019-kb21 L.L.Y & No.lzujbky-2020-kb22 L.Y), and Institute Scientific Research Fund Project/Youth Project(No. 20GSSY4-8 L.L.Y). The funders (Laiyuan Li and Yang Luo) had a role in study design, data collection, analysis, the decision to publish, and preparation of the manuscript.

**Competing interests:** The authors have declared that no competing interests exist.

## Introduction

Neoadjuvant chemoradiotherapy (nCRT) before total mesorectal excision (TME) and followed systemic chemotherapy is widely accepted as the standard therapy for locally advanced rectal cancer (LARC). Although nCRT and preoperative systemic chemotherapy are the preferred approach for LARC, it does not provide better results in terms of disease-free survival (DFS) and overall survival (OS) compared with surgery and adjuvant chemotherapy [1]. The concept of total neoadjuvant therapy (TNT), in which chemotherapy and chemoradiation are administered before TME, can be administered exclusively for patients with widespread LARC [2, 3]. TNT has also become a platform for studying systemic chemotherapy, new radiation sensitizers and immunotherapy agents for LARC [4, 5]. The NCCN guidelines have approved the use of TNT [6], however, the current evidence is preliminary, we tried to further consolidate the evidence through meta-analysis of related studies.

## Methods

### Literature search strategy

A systematic search of medical databases (PubMed, EMBASE and Cochrane Library) was conducted up to the end of July 1, 2021. We used the following search MeSH terms: neoadjuvant therapy, neoadjuvant chemoradiotherapy, preoperative chemoradiotherapy, neoadjuvant chemoradiation, chemoradiotherapy, total neoadjuvant therapy, total mesorectal excision, rectal cancer. We also searched bibliographies of identified reports for additional references.

### Study inclusion and exclusion criteria

Studies inclusion criteria in the meta-analysis were list as follow: (a) randomized controlled trials (RCTs) or non-randomized controlled trials (nRCTs), (b) TNT versus nCRT in LARC, (c) interest outcomes describing the details of pCR, chemoradiotherapy or surgical complications, R0 resection, local recurrence, distant metastasis, DFS and OS. Review/case reports, no comparable data, repeat publications were excluded.

### Data extraction and quality assessment

Two authors performed study selection, evaluation, and data extraction independently, and discrepancies were resolved by consensus including a third author. The primary endpoint was pCR, The secondary endpoints were chemoradiotherapy or surgical complications, R0 resection, local recurrence, distant metastasis, DFS and OS. Study characteristics were extracted independently by two researchers. Corresponding authors were contacted via e-mail for missing data when necessary. The methodological quality of the included studies was assessed using the Jadad Score [total score from 0 (poor) to 5 (excellent)] for RCT or the Newcastle-Ottawa scale (NOS) [total score from 0 (poor) to 9 (excellent)] for nRCTs [7, 8]. A Jadad scale score≥3 points for RCTs and NOS score≥6 points for nRCTs were considered high quality.

### Statistical analysis

Continuous variables were analysed using weighted mean differences (WMD) and 95% confidence intervals (CIs). Pooled odds ratios (ORs) and 95% CIs were calculated for dichotomous variables. Statistical heterogeneity between trials was assessed by using the $I^2$ test, a value of 50% or greater suggests moderate to substantial inconsistency among studies. Random effects models were used if high heterogeneity between study existed, Otherwise, fixed-effects models were used. Subgroup analysis and sensitivity analysis was performed to identify potential heterogeneity [9, 10]. Publication bias was investigated using Begg's and Egger's tests [11]. All

statistical tests were performed using Review Manager Version 5.3 and STATA/SE version 13.1. The significance level was set at p < 0.05.

## Results

Fig 1 shows the selection of relevant studies for inclusion in the meta-analysis. A total of 8 nRCTs with 1502 and 7 RCTs with 2077 enrolled patients enrolled patients were included for this meta-analysis [12–26] (Fig 1), there were 1812 patients in the TNT group and 1767 patients in the nCRT group. The main characteristics of the included studies are summarized in Table 1.

### Primary endpoints

**pCR.**   Overall, including the 15 studies selected, 652 of 3579 patients (18.2%) achieved a pCR after neoadjuvant therapy, 412 of 1812 (22.7%) in the TNT group and 240 of 1767 (13.6%) in the standard group (OR 1.85, 95% CI 1.39–2.46, p < 0.0001), with certain statistical

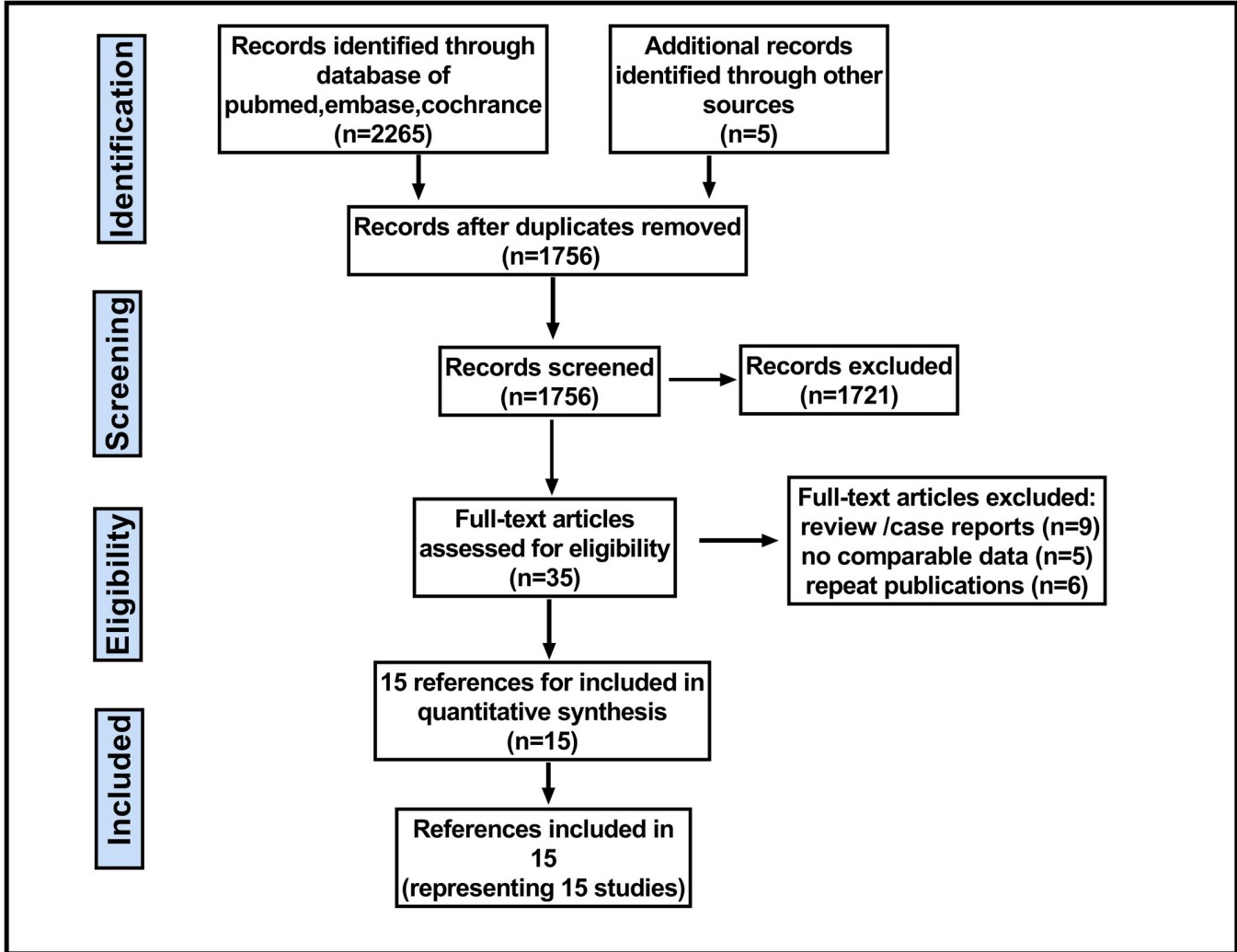

**Fig 1. Flow chart indicating the selection process for this meta-analysis.**

Table 1. Characteristics of the studies included in the meta-analysis.

| First author/ Year | Study design | CRTvsTNT Number of patients | Male gender (%) | Age (year) (mean ± SD) | T4 | N2 | CRM+ | RT Dose (Gys) | TNT regimen | CRT regimen | Follow-up (months) | Study quality |
|---|---|---|---|---|---|---|---|---|---|---|---|---|
| Marechal R/ 2012 | RCT | 29vs28 | 75vs55 | 62.5±8.8vs62 ±14.5 | NR | NR | NR | 45vs45 | FOLFOX * 2→CRT→Sur | CRT(5FU)→Sur | NR | –/3 |
| Fernandez C/ 2015 | RCT | 62vs60 | 70vs65 | 62±8.3vs60±9.5 | 3vs2 | NR | 5vs– | NR | XELOX * 4→CRT→Sur | CRT→Sur→CT | 69 | –/3 |
| Bujko K/2016 | RCT | 254vs261 | 67vs70 | NR | NR | NR | NR | 50.4vs25 | SCRT→FOLFOX * 3→Sur | CRT→ Sur | 35 | –/4 |
| Moore J/2017 | RCT | 24Vs25 | 75vs73 | 60.5 ±12.6 vs59.7 ±9.9 | 163vs165 | NR | NR | NR | CRT→5FU * 3→Sur | CRT→Sur | NR | –/4 |
| Kim SY/2018 | RCT | 55Vs53 | 84vs68 | 55± 8 vs56± 10 | 1vs5 | 19vs15 | 14vs16 | 50.4vs50.4 | CRT→XELOX * 2→Sur | CRT→Sur | NR | 7★/– |
| Conroy T/ 2020 | RCT | 230Vs231 | NR | NR | 10vs9 | NR | NR | 50.4vs50.4 | FOLFIRINOX*6→CRT→Sur | CRT→Sur→ FOLFOX | 46.5 | –/3 |
| Bahadoer RR/ 2021 | RCT | 450vs462 | 69vs65 | 62±2.2vs62±2.3 | 35vs41 | NR | NR | 25vs50.4 | CRT→XELOX *6/→Sur | CRT→Sur→ CT | 54 | –/4 |
| Cui J/2020 | Re | 61vs83 | 56vs69 | 56 ±13.5vs54 ±12 | 20vs15 | NR | NR | NR | CRT→XELOX * 2→Sur | NR | NR | 7★/– |
| Liang HQ/ 2018 | Re | 80vs76 | NR | 64vs75 | 44vs50 | 11vs21 | NR | NR | CRT→mFOLFOX * 2– 4→Sur | CRT→Sur | 32vs30 | 6★/– |
| Cercek A/2018 | Re | 320vs308 | 60vs59 | NR | 57vs62 | 45vs36 | NR | NR | NR | NR | 46vs23 | 7★/– |
| Bhatti AB /2015 | Re | 61vs93 | 69vs68 | NR | 20vs36 | NR | 8vs17* | 50.4vs50.4 | mFOLFOX6 *4→ CRT→Sur | CRT→Sur | NR | 8★/– |
| Thakur N./ 2020 | Pro | 13vs15 | -NR | NR | NR | NR | NR | 25vs25 | CRT→mFOLFOX6 *2→Sur | CRT→Sur | NR | 8★/– |
| van Zoggel D/ 2018 | Pro | 71vs58 | 57Vs79 | 65±9vs64 ±10 | NR | NR | NR | NR | FOLFOX *4→ CRT→Sur | CRT→Sur | NR | 7★/– |
| Garcia-Aguilar J/2015 | Pro | 60vs65 | 62vs63 | 57 ±13.3vs58 ±9.8 | NR | NR | NR | NR | CRT→mFOLFOX6 *6→Sur | CRT→Sur | NR | 7★/– |
| Markovina S/ 2017 | Pro | 69vs69 | 67vs71 | 56.6 ±12.9vs57.2 ±12.6 | 1vs3 | NR | NR | NR | CRT→mFOLFOX6 →Sur | NR | 54.3 vs49.4 | 8★/– |

★ number of stars for Nottingham Ottawa scale for each included trial,* Postradiotherapy, NOS = Nottingham-Ottawa scale, CRT = Chemoradiotherapy, CT = chemotherapy, TNT = total neoadjuvant therapy, Pro = prospective, Sur = surgery; Re = retrospective, RCTs = randomized controlled trial, NR = no report.

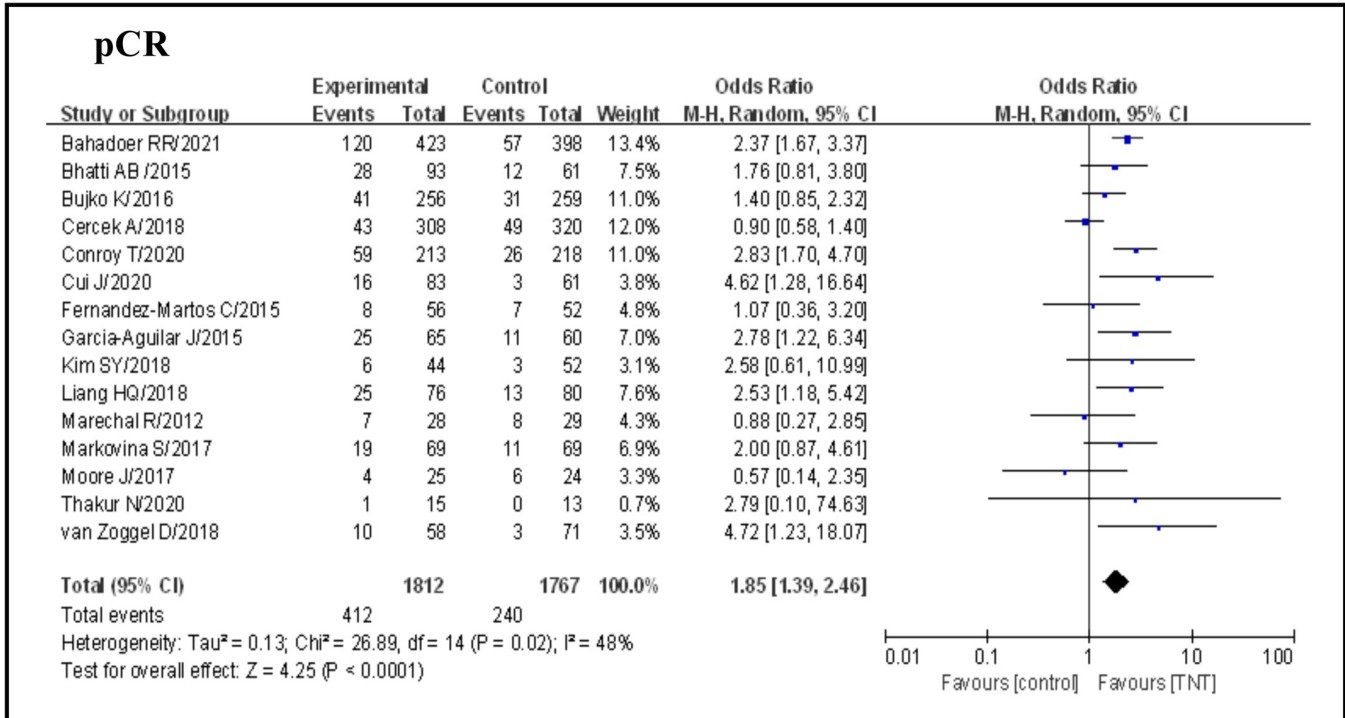

**Fig 2. Forest plot for pCR, OR are shown with 95% confidence interval.**

heterogeneity ($I^2$ = 48%, p = 0.02)(Fig 2). A subgroup analysis showed that single-center or multi-center studies, sample size, RCT or nRCT, neoadjuvant therapy were not causes of heterogeneity in view of the dissimilarity between subgroups (Table 2). The sensibility analysis performed showed stability of the pCR when excluding each study at a time (Table 3).

## Secondary endpoints

**Chemoradiotherapy complications.** Overall, including the 11 studies selected, 1000 of 3534 patients (28.3%) occurred G3–4 adverse events after neoadjuvant therapy, 576 of 1996

**Table 2. Subgroup analyses of PCR based on the study type, setting, sample size and chemoradiotherapy sequence.**

| Outcome | Subgroup | | No. of studies | No. of patients | Study Heterogeneity | | Model | Meta-analysis | |
|---|---|---|---|---|---|---|---|---|---|
| | | | | TNTvsCRT | $I^2$ (%) | P Value | | OR (95%CI) | P Value |
| **pCR** | Study type | RCT | 7 | 1045vs 1032 | 45 | 0.09 | Fixed | 1.75 [1.21, 2.54] | **0.003** |
| | | Prospective | 4 | 207 vs 213 | 0 | 0.76 | Fixed | 2.65 [1.56, 4.50] | **0.0003** |
| | | Retrospective | 4 | 560 vs 522 | 70 | **0.02** | Random | 1.42 [1.03, 1.96] | **0.03** |
| | Setting | Single-center | 7 | 702vs 675 | 55 | **0.04** | Random | 2.01 [1.20, 3.39] | **0.009** |
| | | Multi-center | 8 | 1117vs1095 | 32 | 0.20 | Fixed | 2.19 [1.62, 2.50] | **< 0.00001** |
| | Sample size | ≥200 | 4 | 1200 vs 1195 | 81 | **0.001** | Random | 1.71 [1.02, 2.84] | **0.04** |
| | | < 200 | 11 | 619 vs 565 | 0 | 0.50 | Fixed | 2.24 [1.62, 2.91] | **< 0.00001** |
| | Chemoradiotherapy | CT first | 4 | 498 vs 381 | 31 | 0.19 | Fixed | 2.23 [1.61, 3.27] | **< 0.0001** |
| | | Radiation first | 11 | 1421 vs 1389 | 49 | **0.03** | Random | 1.81 [1.30, 2.51] | **0.0004** |

CRT = Chemoradiotherapy, TNT = total neoadjuvant therapy, OR = odds ratio, CI = confidence interval.

**Table 3. Sensitivity analysis for pathological complete response.**

| Study Excluded | Random Effect | | | Heterogeneity | |
|---|---|---|---|---|---|
| | OR | 95% CI | p Value | I² (%) | I²- P Value |
| van Zoggel D/2018 | 1.79 | 1.34, 2.38 | < 0.00001 | 48 | 0.02 |
| Garcia-Aguilar J/2015 | 1.80 | 1.33, 2.42 | 0.0001 | 50 | 0.02 |
| Markovina S/2017 | 1.84 | 1.36, 2.50 | < 0.0001 | 52 | 0.01 |
| Thakur N/2020 | 1.85 | 1.38, 2.47 | < 0.0001 | 52 | 0.01 |
| Bhatti AB /2015 | 1.86 | 1.37, 2.53 | < 0.0001 | 52 | 0.01 |
| Cercek A/2018 | 2.08 | 1.66, 2.61 | < 0.00001 | 13 | 0.31 |
| Cui J/2020 | 1.79 | 1.34, 2.38 | < 0.0001 | 48 | 0.02 |
| Liang HQ/2018 | 1.81 | 1.33, 2.45 | 0.0001 | 50 | 0.02 |
| Bahadoer RR/2021 | 1.79 | 1.30, 2.45 | 0.0003 | 46 | 0.03 |
| Bujko K/2016 | 1.92 | 1.40, 2.62 | < 0.0001 | 49 | 0.02 |
| Conroy T/2020 | 1.76 | 1.30, 2.37 | 0.0002 | 45 | 0.03 |
| Fernandez-Martos C/2015 | 1.91 | 1.42, 2.56 | < 0.0001 | 50 | 0.02 |
| Kim SY/2018 | 1.83 | 1.36, 2.46 | < 0.0001 | 51 | 0.01 |
| Marechal R/2012 | 1.92 | 1.43, 2.56 | < 0.0001 | 49 | 0.02 |
| Moore J/2017 | 1.92 | 1.45, 2.55 | < 0.00001 | 46 | 0.03 |

OR = odds ratio; CI = confidence interval.

(28.9%) in the TNT group and 424 of 1538 (27.6%) in the standard group (OR 1.03, 95% CI 0.73–1.43, p = 0.88), with certain statistical heterogeneity (I² = 66%, p = 0.001) (Fig 3A).

**Surgical complications.** Overall, including the 10 studies selected, 527 of 2497 patients (21.1%) occurred surgical complications, 267 of 1273 (21.0%) in the TNT group and 260 of 1224 (21.2%) in the standard group (OR 1.02, 95% CI 0.83–1.26, p = 0.83), with no evidence of significant heterogeneity (I² = 0%, p = 0.92) (Fig 3B).

**R0 resection.** Overall, including the 11 studies selected, 1894 of 2238 patients (84.6%) achieved R0 resection, 989 of 1143 (86.5%) in the TNT group and 905 of 1095 (82.6%) in the standard group (OR 1.34, 95% CI 1.05–1.71, p = 0.02), with no evidence of significant heterogeneity (I² = 28%, p = 0.18) (Fig 4A).

**Local recurrence.** Overall, including the two studies selected, 42 of 436 patients (9.6%) occurred local recurrence, 25 of 204 (12.2%) in the TNT group and 17 of 232 (7.3%) in the standard group (OR 1.82, 95% CI 0.95–3.49, p = 0.07), with no evidence of significant heterogeneity (I² = 31%, p = 0.23) (Fig 4B).

**Distant metastasis.** Overall, including the 7 studies selected, 238 of 1080 patients (22.0%) occurred distant metastasis, 111 of 556 (20.0.5%) in the TNT group and 127 of 524 (24.2%) in the standard group (OR 0.77, 95% CI 0.58–1.03, p = 0.08), with no evidence of significant heterogeneity (I² = 38%, p = 0.14) (Fig 4C).

**DFS.** Seven studies reported DFS and the results showed that the TNT group had a significantly longer DFS than the standard group (HR 0.80, 95% CI 0.69–0.92, p = 0.001), with no evidence of significant heterogeneity (I² = 0%, p = 0.43) (Fig 5A).

**OS.** Seven studies reported OS and the results showed that the TNT group had a significantly longer OS than the standard group (HR 0.75, 95% CI 0.62–0.89, p = 0.002), with no evidence of significant heterogeneity (I² = 36%, p = 0.15) (Fig 5B).

**Sensitivity analysis and publication bias.** The sensibility analysis performed showed stability of the pooled OR with pCR when excluding each study at a time. There was no

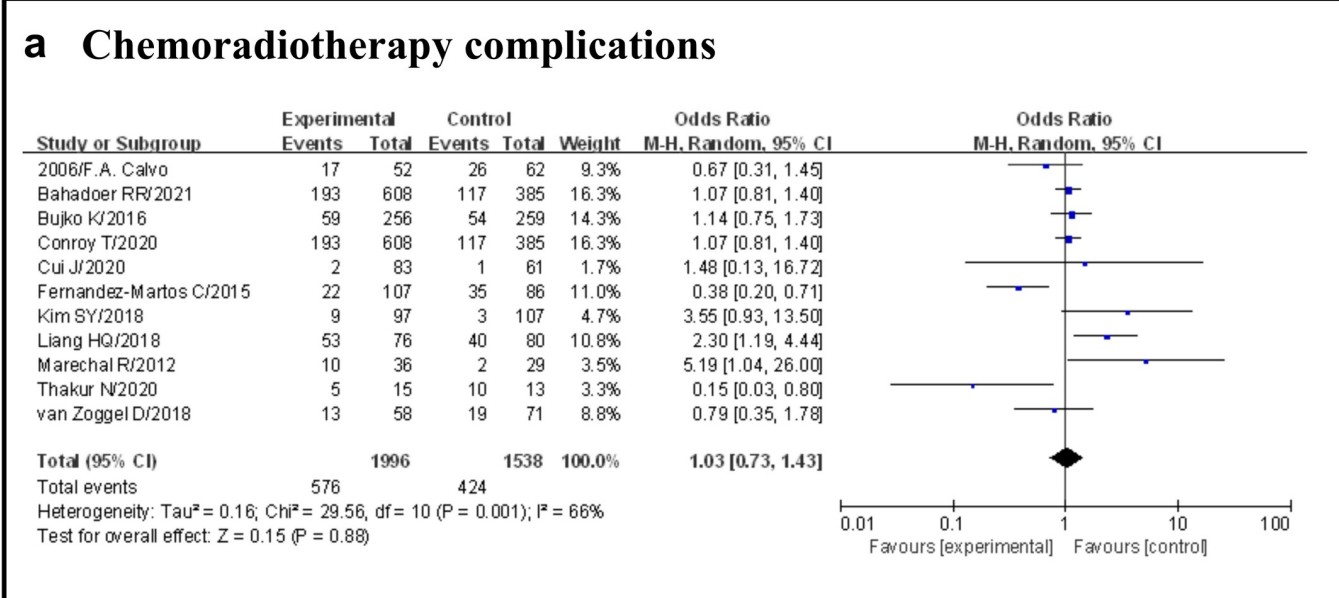

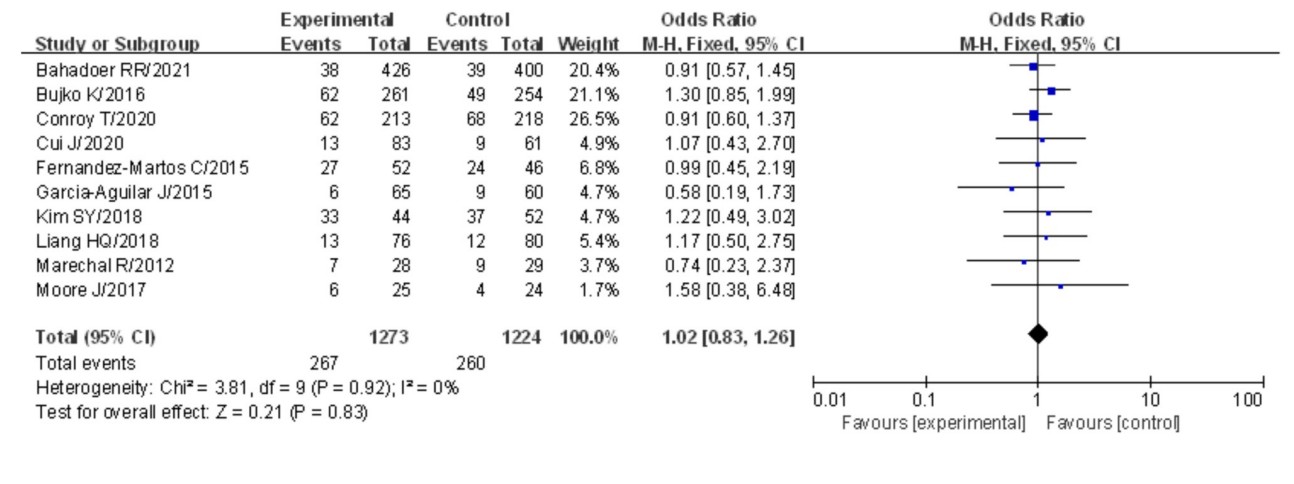

**Fig 3. Forest plot for chemoradiotherapy complications and surgical complications, OR are shown with 95% CIs.**

significant publication bias in the analyses of pCR (Egger's test, p = 0.850; Begg's test, p = 0.729) (Fig 6).

## Discussion

In the meta-analysis, we compared the efficacy of TNT with that of standard therapy for LARC. TNT appears to have advantages over standard therapy for LARC in terms of pCR, R0 resection, DFS, and OS, with comparable nCRT and postoperative complications, and no increase in local recurrence and distant metastasis.

In this meta-analysis, we observed that the overall pCR rate of TNT may increase compared with chemoradiotherapy. Our results were consistent with previous reports [27]. However,

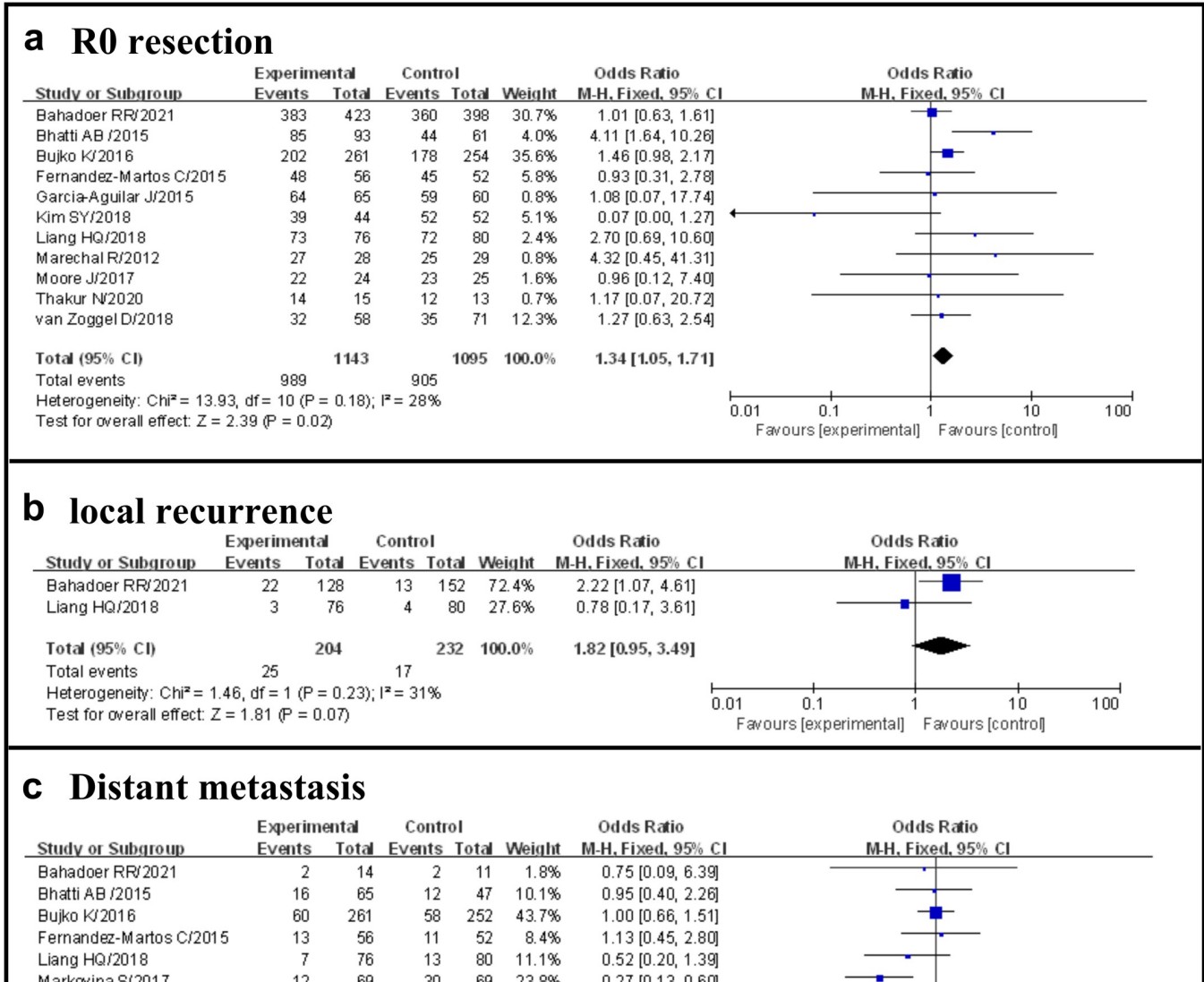

**Fig 4. Forest plot for R0 resection and distant metastasis, OR are shown with 95% Cis.**

our meta-analysis included more and newer studies, which further confirmed the advantage of TNT in rectal cancer. In the review of 16 studies involving a total of 3363 patients, patients who experienced pCR had a 75%, lower risk of distant (8.7%) and local (0.7%) recurrence than patients who did not [28]. Compared with patients with incomplete response, patients with pCR after neoadjuvant therapy are less likely to have local recurrence and more likely to have better survival outcomes [29]. pCR is considered to be the key prognostic criterion for the long-term prognosis of LARC [30]. The use of TNT can significantly reduce tumor volume,

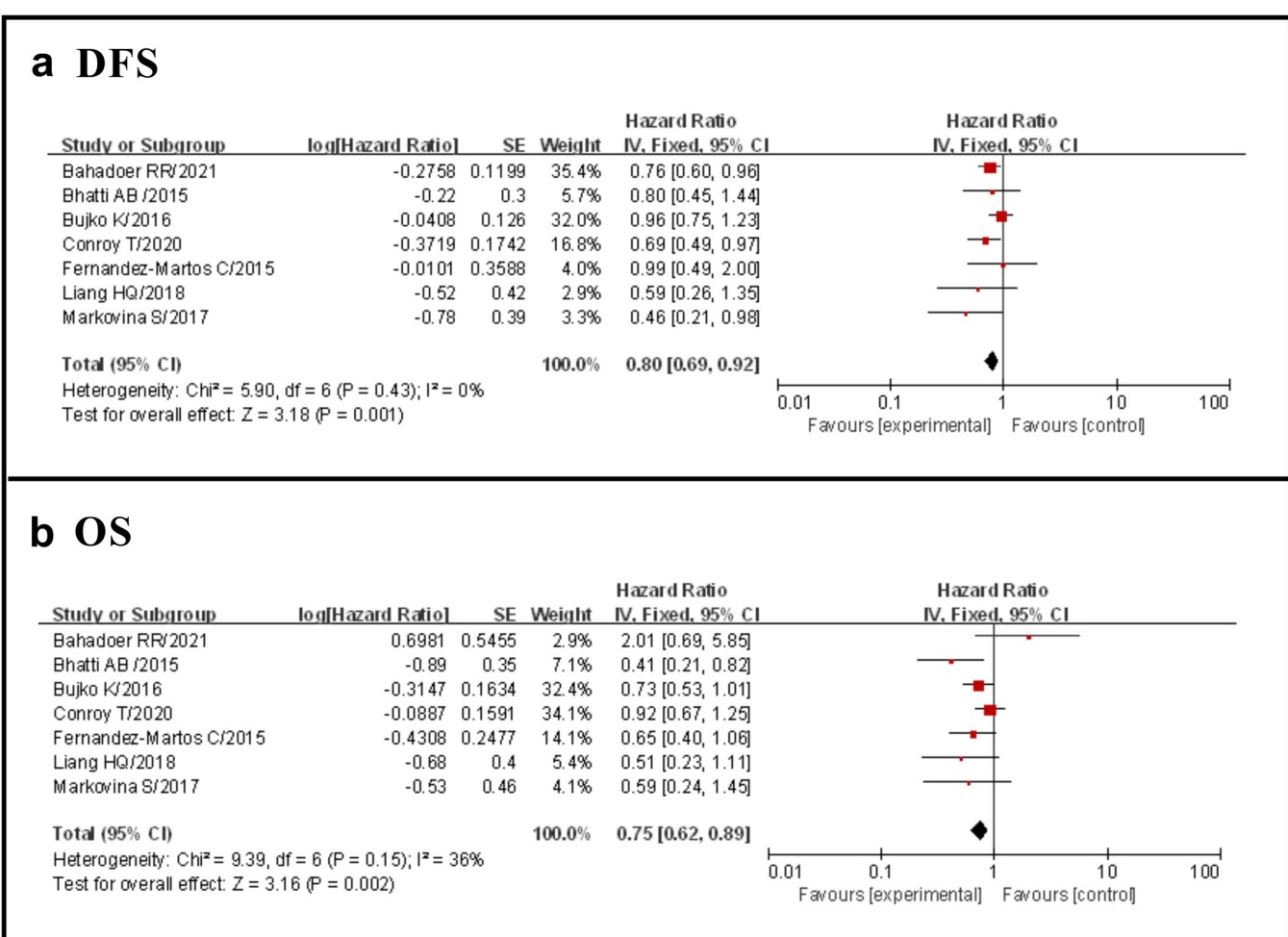

**Fig 5. Forest plot for DFS and OS, HR are shown with 95% Cis.**

which may lead to more patients adopting non-surgical observation and waiting strategies in the future. Habr-Gama et al introduced the watch-and-wait strategy [31], they reported that after 10 years of follow-up, patients who gave up surgery after obtaining complete clinical response had an OS rate of 97.7% and a DFS rate of 84%. pCR for rectal cancer is associated with excellent long-term prognosis, independent of treatment strategy. Surgical resection may not improve outcomes while increasing the incidence of temporary or permanent stoma and unnecessary morbidity and mortality. Another potential advantage of simultaneous chemotherapy and radiochemotherapy is that it can avoid or delay surgery when a complete clinical response is observed. The watch-and-wait approach may be considered a better treatment strategy, because surgery may lead to intestinal or bladder incontinence and sexual dysfunction, as well as short-term or permanent stoma [32–35].

Adjuvant chemotherapy is recommended by current guidelines, however, patient compliance is poor and survival benefit is unclear. Although the nCRT approach has made significant improvements in local control, the incidence of distal metastasis has not decreased. In our meta-analysis, the rate of DFS and OS was significantly higher in the TNT group than in the nCRT group, although studies have shown that watch-and-wait approach is associated with higher rectal preservation, which may be at the expense of lower OS rates

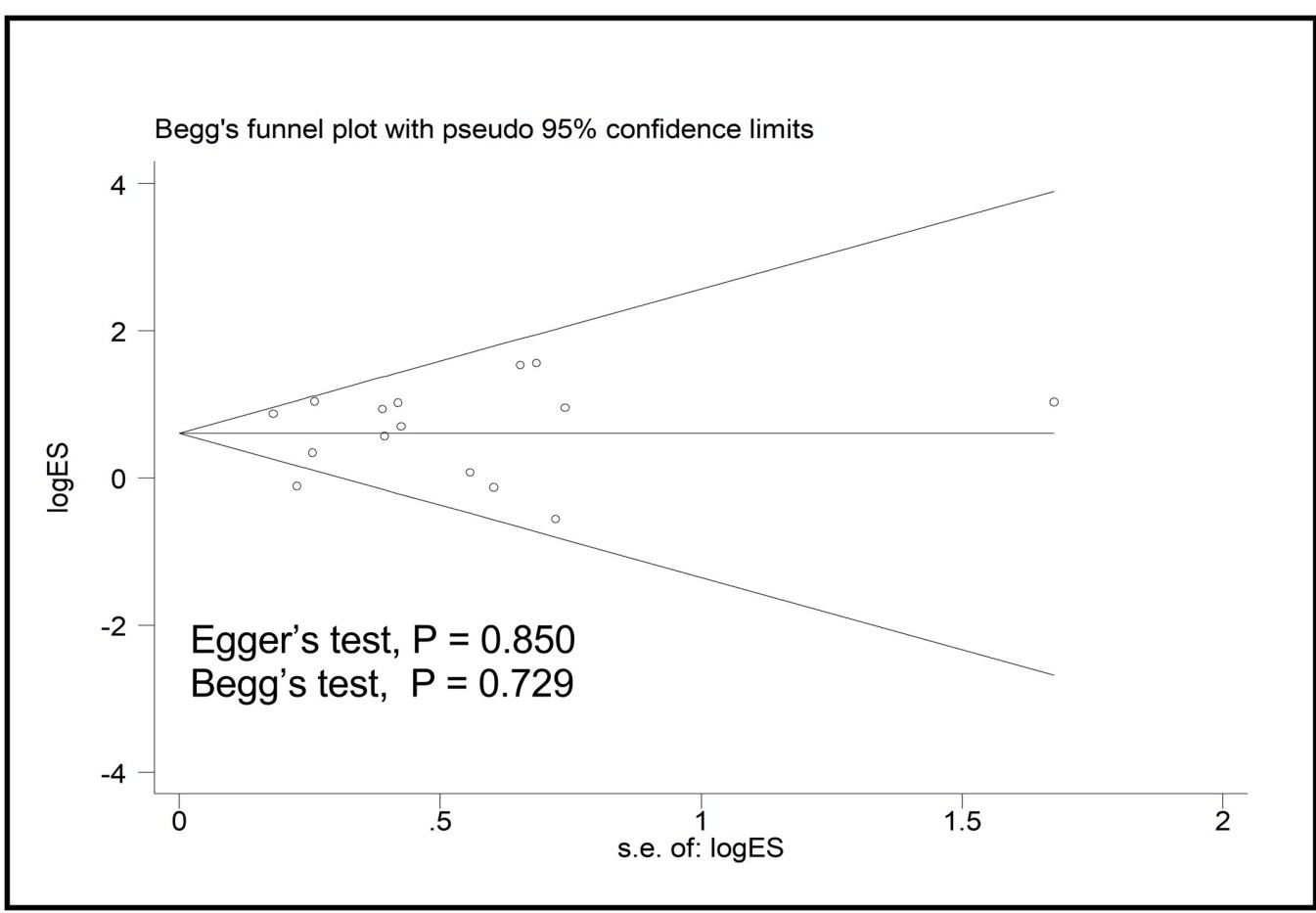

**Fig 6. Funnel plot using Begg method for pCR.**

and increased risk of distant metastasis [36]. In addition, treatment-related toxicity appeared to be acceptable in all patients who completed TNT therapy. Although TNT was associated with an increased incidence of pelvic fibrosis and a longer interval from nCRT to surgery, TNT did not lead to an increase in surgical difficulty or postoperative complications [17]. However, these results should be interpreted with caution, as the lack of standardization of postoperative morbidity reports limits the interpretation of the results. Another disadvantage is the short follow-up period, so the assessment of long-term distant metastases needs to be cautious.

Assessment of quality of life (QOL) was not included in this meta-analysis, in the EXPERT-C trial, intensified neoadjuvant strategy for QoL and bowel function did not appear to be significantly affected [37]. This study was conducted at an appropriate time, because enough data was available to use meta-analytical methods which enabled us to provide the most up-to-date information on this topic.

Our study has the following limitations. First, although the meta-analysis including RCT is ideal, the limited number of RCTs prevents us from drawing clear conclusions. Second, the follow-up included in this meta-analysis is short, and the long-term survival benefit remains to be confirmed. However, this meta-analysis was completed at the appropriate time, and we provide the latest information in this area.

In conclusion, TNT appears to have advantages over standard therapy for LARC in terms of pCR, R0 resection, DFS, and OS, with comparable nCRT and postoperative complications, and no increase in local recurrence and distant metastasis. If these findings can be applied clinically, more patients with LARC will be eligible for organ preservation, which will avoid surgical sequelae and improve quality of life.

## Supporting information

**S1 Checklist. PRISMA 2009 checklist.**
(DOC)

## Author Contributions

**Methodology:** Shujie Ma, Yinyin Guo.

**Resources:** Xiping Zhang.

**Software:** Yang Luo.

**Writing – original draft:** Laiyuan Li.

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
