## [Decision Letter · Decision Letter 0]

16 Jun 2022

PONE-D-22-09905Total Neoadjuvant Therapy versus Standard Therapy in Locally Advanced Rectal Cancer:A Systematic Review and Meta-Analysis of 16 TrialsPLOS ONE

Dear Dr. Li,

Thank you for submitting your manuscript to PLOS ONE. After careful consideration, we feel that it has merit but does not fully meet PLOS ONE’s publication criteria as it currently stands. Therefore, we invite you to submit a revised version of the manuscript that addresses the points raised during the review process.

We look forward to receiving your revised manuscript.

Kind regards,

Norikatsu Miyoshi, M.D., Ph.D., FACS

Academic Editor

PLOS ONE

Journal Requirements:

“This work was supported by the Natural Science Foundation of Gansu Province (No.20JR10RA371, L.LY), Fundamental Research Funds for the Central Universities (No.lzujbky-2019-kb21 L.L.Y & No.lzujbky-2020-kb22 L.Y), and Institute Scientific Research Fund Project/Youth Project(No. 20GSSY4-8 L.L.Y).”

“The Natural Science Foundation of Gansu Province (No.20JR10RA371, L.LY), Fundamental Research Funds for the Central Universities (No.lzujbky-2019-kb21 L.L.Y & No.lzujbky-2020-kb22 L.Y), and Institute Scientific Research Fund Project/Youth Project(No. 20GSSY4-8 L.L.Y).”

3. Please amend the manuscript submission data (via Edit Submission) to include author “Xiping Zhang, Shujie Ma, Yinyin Guo, Yang Luo”

4. We note that this manuscript is a systematic review or meta-analysis; our author guidelines therefore require that you use PRISMA guidance to help improve reporting quality of this type of study. Please upload copies of the completed PRISMA checklist as Supporting Information with a file name “PRISMA checklist”

6. Thank you for stating the following financial disclosure:

 “The Natural Science Foundation of Gansu Province (No.20JR10RA371, L.LY), Fundamental Research Funds for the Central Universities (No.lzujbky-2019-kb21 L.L.Y & No.lzujbky-2020-kb22 L.Y), and Institute Scientific Research Fund Project/Youth Project(No. 20GSSY4-8 L.L.Y).”

Reviewers' comments:

Reviewer's Responses to Questions

**Comments to the Author**

1. Is the manuscript technically sound, and do the data support the conclusions?

Reviewer #1: Yes

Reviewer #2: Yes

2. Has the statistical analysis been performed appropriately and rigorously? 

Reviewer #1: Yes

Reviewer #2: Yes

3. Have the authors made all data underlying the findings in their manuscript fully available?

Reviewer #1: Yes

Reviewer #2: Yes

4. Is the manuscript presented in an intelligible fashion and written in standard English?

Reviewer #1: Yes

Reviewer #2: Yes

5. Review Comments to the Author

Reviewer #1: These findings come from a meta-analysis of 16 research, eight of which were non-randomized and eight of which were randomized controlled trials comparing TNT to regular CRT.

However, there are a few aspects that need to be clarified:

1. Before therapy, the characteristics of tumors in both groups should be evaluated and the results given (T stage N stage, whether they threaten CRM).

2. Both groups' post-radiotherapy waiting periods must be compared.

3. Distant metastases were investigated, but not local recurrence rates. Local recurrence rates, which are a crucial predictor of preoperative neoadjuvant therapy success, should not be overlooked.

4. Minor correction: Disease-free survival should be shortened to DFS.

Reviewer #2: The authors meta-analysis answered a significant research question in the field of surgical oncological treatment for locally advanced rectal cancer . However, the following issues needed to be addressed before this piece of work could be published

1. There are spelling mistakes "PFS wrongly used for DFS"

2. The authors didn't give a based argument on the subject of TNT and standard care (CRT). In line 5, under discussion, the authors were silent /did not discuss the controversies surrounding the result from Habr -Gama et al, there are other excellent centres in the world who have not been able to reproduce the same result following TNT

3. The sentence in 15 is unclear to me "In this meta-analysis, the DFS and distant metastasis of TNT were significantly superior to nCRT alone, despite studies have shown that the watch-and-wait approach is associated with higher preserved rectum, which may be at the cost of reducing the rate of OS and increasing the risk of distant metastasis"

4. There is a similar publication on this subject by Liu S, Jiang T, Xiao L, Yang S et .at in cancer 2021. Why would you repeat a similar meta-analysis within a year part. I would have expected the authors to make reference to this study and show us a clear benefit of there's over the 2021 publication- this have not been done.

5. I would have expected the authors to buttress the positive findings from TNT by citing other authors report and perhaps proffer suggestion why there is no significant difference as regards surgical complication , R0 resection rate and distant metastasis .

6. I am not sure if the authors were meant to use the term superiority when referencing to the overall benefit of TNT over standard CRT. The superiority and inferiority test mostly applies to original research where is a null or alternative hypothesis to prove or disprove , hence to my understanding it does not apply to this meta-analysis.

7. There are other limitations to this study that was not discussed.

8. Would have expected the authors to make reference to QOL in TNT vs CRT even though , this might have not been discussed in the other main articles.

6. PLOS authors have the option to publish the peer review history of their article (what does this mean?). If published, this will include your full peer review and any attached files.

Reviewer #1: **Yes: **Cihangir Akyol,MD.

Reviewer #2: No

---

## [Author Response · Author response to Decision Letter 0]

21 Jul 2022

Dear Editors and Reviewers:

Thank you for your letter and for the reviewers’ comments concerning our manuscript entitled “Total Neoadjuvant Therapy versus Standard Therapy in Locally Advanced Rectal Cancer:A Systematic Review and Meta-Analysis of 15 Trials” (ID: PONE-D-22-09905).Those comments are all valuable and very helpful for revising and improving our paper, as well as the important guiding significance to our researches. We have studied comments carefully and have made correction which we hope meet with approval. Revised portion are marked in red in the paper. The main corrections in the paper and the responds to the reviewer’s comments are as flowing:

We appreciate for Editors/Reviewers’ warm work earnestly, and hope that the correction will meet with approval.

Once again, thank you very much for your comments and suggestions.

 Laiyuan Li

Review Comments to the Author

Reviewer #1: These findings come from a meta-analysis of 16 research, eight of which were non-randomized and eight of which were randomized controlled trials comparing TNT to regular CRT.

However, there are a few aspects that need to be clarified:

1.Before therapy, the characteristics of tumors in both groups should be evaluated and the results given (T stage N stage, whether they threaten CRM).

Response –We are very sorry for our negligence of characteristics of tumors, We have added CRM+ in Table 1, due to space constraints, only the stage of T4 N2 was added.

2.Both groups' post-radiotherapy waiting periods must be compared.

Response –We have added in table 1.

3.Distant metastases were investigated, but not local recurrence rates. Local recurrence rates, which are a crucial predictor of preoperative neoadjuvant therapy success, should not be overlooked.

Response –We have added it in methods and results section.

4. Minor correction: Disease-free survival should be shortened to DFS.

Response –Yes, we have revised it.Special thanks to you for your careful revision. 

Reviewer #2: The authors meta-analysis answered a significant research question in the field of surgical oncological treatment for locally advanced rectal cancer . However, the following issues needed to be addressed before this piece of work could be published

1.There are spelling mistakes "PFS wrongly used for DFS"

Response –We are very sorry for our incorrect writing, we have revised it.

2.The authors didn't give a based argument on the subject of TNT and standard care (CRT). In line 5, under discussion, the authors were silent /did not discuss the controversies surrounding the result from Habr -Gama et al, there are other excellent centres in the world who have not been able to reproduce the same result following TNT

Response –We have made correction according to the comments.We made the following comments about the study: pCR for rectal cancer is associated with excellent long-term prognosis, independent of treatment strategy. Surgical resection may not improve outcomes while increasing the incidence of temporary or permanent stoma and unnecessary morbidity and mortality.Please find it on page 6, line 15.

3.The sentence in 15 is unclear to me "In this meta-analysis, the DFS and distant metastasis of TNT were significantly superior to nCRT alone, despite studies have shown that the watch-and-wait approach is associated with higher preserved rectum, which may be at the cost of reducing the rate of OS and increasing the risk of distant metastasis"

Response –Considering the your suggestion,we have revised it as follow: In our meta-analysis, the rates of DFS and OS was significantly higher in the TNT group than in the nCRT group, although studies have shown that watch-and-wait approach is associated with higher rectal preservation, which may be at the expense of lower OS rates and increased risk of distant metastasis.Please find it on page 6, line 27.

4.There is a similar publication on this subject by Liu S, Jiang T, Xiao L, Yang S et .at in cancer 2021. Why would you repeat a similar meta-analysis within a year part. I would have expected the authors to make reference to this study and show us a clear benefit of there's over the 2021 publication- this have not been done.

Response –Our comments are follow: Our results were consistent with previous reports.(PMID 33987952), however, our meta-analysis included more and newer studies, which further confirmed the advantage of TNT in rectal cancer. This study was conducted at an appropriate time, because enough data was available to use meta-analytical methods which enabled us to provide the most up-to-date information on this topic.Please find it on page 6, line 3.

Liu S, Jiang T, Xiao L, et al. Total Neoadjuvant Therapy (TNT) versus Standard Neoadjuvant Chemoradiotherapy for Locally Advanced Rectal Cancer: A Systematic Review and Meta-Analysis. The oncologist 2021.

5.I would have expected the authors to buttress the positive findings from TNT by citing other authors report and perhaps proffer suggestion why there is no significant difference as regards surgical complication , R0 resection rate and distant metastasis.

Response –Special thanks to you for your good comments.We have added relevant discussion as follow: However, these results should be interpreted with caution, as the lack of standardization of postoperative morbidity reports limits the interpretation of the results.Another disadvantage is the short follow-up period, so the assessment of long-term distant metastases needs to be cautious. Please find it on page 7, line 4.

6.I am not sure if the authors were meant to use the term superiority when referencing to the overall benefit of TNT over standard CRT. The superiority and inferiority test mostly applies to original research where is a null or alternative hypothesis to prove or disprove , hence to my understanding it does not apply to this meta-analysis.

Response –We have re-written this sentence according to your suggestion:The rates of pCR was significantly higher in the TNT group than in the nCRT group.Please find it on page 1, line 25.

7.There are other limitations to this study that was not discussed.

Response –Considering the Reviewer’s suggestion, we have added discussion as follow: However, these results should be interpreted with caution, as the lack of standardization of postoperative morbidity reports limits the interpretation of the results.Another disadvantage is the short follow-up period, so the assessment of long-term distant metastases needs to be cautious.Please find it on page 6, line 4.

8.Would have expected the authors to make reference to QOL in TNT vs CRT even though , this might have not been discussed in the other main articles.

Response –We are very sorry for our incorrect writing, we have added it: Assessment of QoL was not included in this study, in a recent analysis of the EXPERT-C trial, intensified neoadjuvant strategy for QoL and bowel function did not appear to be significantly affected (PMID:26031368). Nevertheless, the absence of a control group of standard CRT limits the general applicability of our results and a definitive conclusion on the role of TNT in high-risk LARC can only be provided by randomised phase III trials.Please find it on page 7, line 9.

Sclafani F, Peckitt C, Cunningham D, et al. Short- and Long-Term Quality of Life and Bowel Function in Patients With MRI-Defined, High-Risk, Locally Advanced Rectal Cancer Treated With an Intensified Neoadjuvant Strategy in the Randomized Phase 2 EXPERT-C Trial. Int J Radiat Oncol Biol Phys 2015; 93(2): 303-12.

6. PLOS authors have the option to publish the peer review history of their article (what does this mean?). If published, this will include your full peer review and any attached files.

Response –I agree with including full peer review and any attached files.Special thanks to you for your good comments.

---

## [Decision Letter · Decision Letter 1]

11 Oct 2022

Total Neoadjuvant Therapy versus Standard Therapy in Locally Advanced Rectal Cancer:A Systematic Review and Meta-Analysis of 15 Trials

PONE-D-22-09905R1

Dear Dr. Li,

We’re pleased to inform you that your manuscript has been judged scientifically suitable for publication and will be formally accepted for publication once it meets all outstanding technical requirements.

Kind regards,

Alessandro Rizzo

Academic Editor

PLOS ONE

Additional Editor Comments (optional):

Reviewers' comments:

Reviewer's Responses to Questions

**Comments to the Author**

1. If the authors have adequately addressed your comments raised in a previous round of review and you feel that this manuscript is now acceptable for publication, you may indicate that here to bypass the “Comments to the Author” section, enter your conflict of interest statement in the “Confidential to Editor” section, and submit your "Accept" recommendation.

Reviewer #1: All comments have been addressed

2. Is the manuscript technically sound, and do the data support the conclusions?

Reviewer #1: Yes

3. Has the statistical analysis been performed appropriately and rigorously? 

Reviewer #1: Yes

4. Have the authors made all data underlying the findings in their manuscript fully available?

Reviewer #1: Yes

5. Is the manuscript presented in an intelligible fashion and written in standard English?

Reviewer #1: Yes

6. Review Comments to the Author

Reviewer #1: I would want to express my gratitude to the authors for all of the revisions. I believe that this study will be published in the form of a meta-analysis in the scientific literature. This meta-analysis will be an important cornerstone in the multidisciplinary treatment of rectal cancer, which is updated every day.

I would also like to thank them for detecting the duplicate article that both reviewers overlooked and removed it from the study after they realized there was a problem.

My opinion is that the version of the paper that is now being worked on is ready for publication.

7. PLOS authors have the option to publish the peer review history of their article (what does this mean?). If published, this will include your full peer review and any attached files.

Reviewer #1: **Yes: **Cihangir Akyol

---

## [Editor Report · Acceptance letter]

13 Oct 2022

PONE-D-22-09905R1 

Total Neoadjuvant Therapy versus Standard Therapy in Locally Advanced Rectal Cancer:A Systematic Review and Meta-Analysis of 15 Trials 

Dear Dr. Li:

I'm pleased to inform you that your manuscript has been deemed suitable for publication in PLOS ONE. Congratulations! Your manuscript is now with our production department. 

Kind regards, 

on behalf of

Dr. Alessandro Rizzo 

Academic Editor

PLOS ONE